# Genome-Wide Analysis of DREB Family Genes and Characterization of Cold Stress Responses in the Woody Plant *Prunus nana*

**DOI:** 10.3390/genes14040811

**Published:** 2023-03-28

**Authors:** Cheng Qian, Lulu Li, Huanhuan Guo, Gaopu Zhu, Ning Yang, Xiaoyan Tan, Han Zhao

**Affiliations:** 1College of Landscape Architecture and Forestry, Qingdao Agricultural University, Qingdao 266109, China; 2Zhengzhou Botanical Garden, Zhengzhou 450042, China; 3Research Institute of Non-Timber Forestry, Chinese Academy of Forestry, Zhengzhou 450014, China; 4Qingdao Landscape and Forestry Integrated Service Center, Qingdao 266003, China

**Keywords:** *Prunus nana*, wild almond, gene family

## Abstract

Dehydration response element binding factor (DREB) is a family of plant-specific transcription factors, whose members participate in the regulation of plant responses to various abiotic stresses. *Prunus nana*, also known as the wild almond, is a member of the *Rosaceae* family that is rare and found to grow in the wild in China. These wild almond trees are found in hilly regions in northern Xinjiang, and exhibit greater drought and cold stress resistance than cultivated almond varieties. However, the response of *P. nana DREB*s (*PnaDREB*s) under low temperature stress is still unclear. In this study, 46 *DREB* genes were identified in the wild almond genome, with this number being slightly lower than that in the sweet almond (*Prunus dulcis* cultivar ‘Nonpareil’). These *DREB* genes in wild almond were separated into two classes. All *PnaDREB* genes were located on six chromosomes. *PnaDREB* proteins that were classified in the same groups contained specific shared motifs, and promoter analyses revealed that *PnaDREB* genes harbored a range of stress-responsive elements associated with drought, low-temperature stress, light responsivity, and hormone-responsive cis-regulatory elements within their promoter regions. MicroRNA target site prediction analyses also suggested that 79 miRNAs may regulate the expression of 40 of these *PnaDREB* genes, with *PnaDREB2*. To examine if these identified *PnaDREB* genes responded to low temperature stress, 15 of these genes were selected including seven homologous to *Arabidopsis* C-repeat binding factor (*CBF*s), and their expression was assessed following incubation for 2 h at 25 °C, 5 °C, 0 °C, −5 °C, or −10 °C. In summary, this analysis provides an overview of the *P. nana PnaDREB* gene family and provides a foundation for further studies of the ability of different *PnaDREB* genes to regulate cold stress responses in almond plants.

## 1. Introduction

Plant growth can be profoundly regulated by adverse environmental factors including drought and cold stress [1,2]. Adverse environmental conditions drive plants to upregulate specific TFs and other genes that help mitigate these stressors or their impacts on the host plant. The AP2/ERF (APETALA2/ethylene response factor) superfamily is the largest group of TFs in plants, regulating their development and ability to tolerate a wide array of abiotic and biotic stressors [3,4]. According to amino acid sequence similarity and the number of conserved domains, the AP2/ERF family is classified into AP2, ERF, DREB, RAV, and five other subgroups. The DREB and ERF subgroups only encode one AP2 conserved domain [5,6,7,8]. Dehydration response element binding factor (DREB) proteins are members of the ERF subfamily [9], each of which harbors conserved AP2/ERF domains that bind to the dehydration-responsive element (DRE) regions of the DNA with a core motif of A/GCC GAC [10,11,12]. The domain is composed of 60~70 amino acids, and the encoded protein is mainly composed of α-spiral, and β-folding is primary [13,14]. DREB has many special biological functions due to its unique domain. Members of this AP2/ERF TF family have been widely studied in many plant species, and have been shown to be closely associated with cold stress responses [15]. A range of studies have shown that the cold tolerance of transgenic *Arabidopsis* or tobacco plants can be enhanced through the overexpression of homologous *DREB* genes from wheat, maize, and rice [16,17]. The *CBF1*/*DREB1B*, *CBF2*/*DREB1C*, and *CBF3*/*DREB1A* cold-inducible genes are involved in regulating *Arabidopsis* responses to low-temperature conditions [18].

DREB family proteins have been characterized in a diverse range of plant species including *Arabidopsis thaliana* [19,20] and *Oryza sativa* [7]. DREB family proteins encoded by *A. thaliana* are classified into six groups numbered A-1 through A-6 [21]. Members of the A-1 DREB protein family are associated with regulating cold stress responses, and similar genes in *O. sativa* including *OsDREB1A*, *OsDREB1B*, and *OsDREB1C* were found to augment the ability of plants to tolerate cold conditions through binding to GCC box elements [21]. The DREB A-2 protein family consists of proteins related to tolerance to drought and saline conditions [20]. The A-3 subfamily protein ABI4 is reportedly linked to sugar- and ABA-related signaling [9], while the *Zea mays* A-4 subfamily member *ZmDREB4.1* has been associated with negatively regulating the growth and development of these plants [22]. The A-5 subfamily protein *ScDREB8* identified in *Syntrichia caninervis* enhanced the ability of *A. thaliana* to tolerate saline stress via the upregulation of a variety of stress-associated genes [23]. The A-6 protein *CmDREB6* identified in *Chrysanthemum morifolium* can, when overexpressed, enhance the ability of plants to tolerate heat stress conditions [24].

CBF (C repeat binding factor) is a class of transcription factors related to plant stress, which plays a key regulatory role in plant non-stress response [4,25]. This transcription factor was first discovered and identified in *Arabidopsis*. The expression of *CBF*s is regulated by upstream factors including the ICE1 TF [26], which is upregulated rapidly following exposure to cold conditions [27]. The rapid induction of *CBF1*/*DREB1B* gene within minutes of low-temperature exposure is followed by downstream CRT/DRE element-containing gene expression. *CBF1*/*DREB1B* has also been definitively shown to protect plants against cold stress [28], with strawberry plants expressing the *Arabidopsis CBF1*/*DREB1B* gene exhibiting enhanced low-temperature stress resistance [29]. *CBF3*/*DREB1A* can similarly augment *Arabidopsis* cold resistance when overexpressed [30]. *CBF1*/*DREB1B* and *CBF3*/*DREB1A* are expressed at earlier time points than *CBF2*/*DREB1C* under cold conditions where upon they promote *CBF2*/*DREB1C* upregulation, which in turn provided inhibitory feedback to suppress *CBF1*/*DREB1B* and *CBF3*/*DREB1A* expression [31]. *CBF4*/*DREB1D* induction is not responsive to cold temperatures, but it is nonetheless able to improve the ability of *Arabidopsis* plants to resist the cold [32]. Prior studies of *Arabidopsis DREB*s and the phylogenetic tree results generated herein were used to select 15 *PnaDREB* genes from *P. nana* for qPCR-based validation of their responsiveness to cold stress.

*P. nana*, also known as the wild almond tree, is an evolutionarily ancient species that belongs to the *Amygdalus* subgenus in the *Rosaceae* family. It can be found growing in southeastern Europe and central/western Asia, growing only at an elevation of 900–1200 m in the hilly regions of northwest Xingjiang in China. This perennial species grows rapidly but is endangered, so is classified as a national key protected wild tree species in China. It exhibits high levels of drought and cold tolerance, and is valued as an ornamental species, with its seeds also being used for oil extraction, consumption, or pharmaceutical applications [33]. These wild almond species also represent an invaluable genetic resource with the potential to enhance the stress tolerance of cultivated almond trees through targeted breeding or selection efforts. However, due to biological/abiotic stress, the production of *P. nana* is limited and the promotion efforts are insufficient. Moreover, no studies to date have characterized the DREB protein family in *P. nana*. *DREB* genes play an important role in plant development and response to biotic and abiotic stresses, which means that these genes may have the potential to be used to improve *P. nana* production. Accordingly, the whole genome analysis of *DREB* family genes in wild almond plants was herein performed, with further analyses being conducted to identify candidate *DREB* family TFs that may be associated with cold tolerance in this species. Therefore, understanding the cold response mechanism of *P. nana* is an important way to develop the *P. nana* industrial chain, screen high-quality genotypes of cold tolerant plants, and breeding.

In this paper, we excavated 46 *DREB* genes from the *P. nana* genome. Their conserved motifs, intron/exon structures, domains, interaction analyses, and cis-acting regulatory elements were systematically analyzed. Meanwhile, their evolutionary relationships with dicotyledonous *A. thaliana*, *P. dulcis* cultivar ‘Nonpareil’, *P. avium*, *P. mume*, *P. persica*, and *P. armeniaca* were compared, and further analyses were performed to gauge their possible roles in the regulation of environmental stress responses. Overall, these results provide a basis for further studies of the role of *DREB* family genes in the genetic improvement of stress resistance in cultivated almonds, and aimed to provide convenience for further analysis of the functions of the *DREB* genes in *P. nana*.

## 2. Materials and Methods

### 2.1. DREB Family Transcription Factor Identification

A combined search strategy was employed to detect *DREB* transcription factors within the *P. nana* genome. Initially, a hidden Markov model (HMM) file corresponding to AP2 (PF00847) was utilized as a query to search the genomes of cultivated and wild almond plants, with the later dataset having been unpublished. Of the identified proteins harboring AP2/ERF domains, proteins in the *DREB* subfamily were selected based upon the detection of conserved amino acids at positions 14 and 19 within this AP2/ERF domain. These protein sequences were further assessed to validate these AP2/ERF domains through an NCBI CD-search query [34]. DREB protein classification was achieved in other plants (*A. thaliana*, *P. dulcis* cultivar ‘Nonpareil’, *P. avium*, *P. mume*, *P. persica*, and *P. armeniaca*) by using this same approach with *P. nana*. After which, MEGA6 was used to construct a maximum likelihood (ML) phylogenetic tree based on full-length DREB protein sequences that had been aligned using Clustalx2.1, with 500 bootstrap replicates.

Duplicate *PnaDREB* sequences were identified by collinearity analysis with MCScanX software [35]. Nonsynonymous (Ka) and synonymous (Ks) *PnaDREB* sequence substitutions were estimated with DnaSP in order to predict the evolutionary strain (Ka/Ks) and divergence time [36].

### 2.2. Plant Cultivation and Cold Stress Treatment

A total of 15 genes were selected for the plant culture and cold stress treatment experiments, of which *PnaDREB22*, *PnaDREB31*, *PnaDREB36*, *PnaDREB38*, *PnaDREB40*, *PnaDREB41*, *PnaDREB42*, and *PnaDREB43* were homologous genes to CBFs, while *PnaDREB1*, *PnaDREB3*, *PnaDREB9*, *PnaDREB12*, *PnaDREB16*, *PnaDREB18*, *PnaDREB19*, and *PnaDREB27* were selected as low temperature homeopathic response elements. Wild (*P. dulcis*) and sweet almond (*P. dulcis* cultivar ‘Nonpareil’) seeds were treated for 24 h with a solution containing gibberellin (200 mM) prior to planting in a culture chamber. Seeds were then cultivated under consistent conditions (22 °C, 16 h light/8 h dark). Gene expression changes in response to different levels of cold stress were evaluated by transferring 2-month-old *P. nana* seedlings into incubators and then cooling them at a rate of 3°/h to the final target temperature (25 °C, 5 °C, 0 °C, −5 °C, and −10 °C), which was held for 2 h. Leaves were then harvested from these seedlings, snap-frozen with liquid nitrogen, and stored at −80 °C.

### 2.3. qPCR Analyses

Total RNA was extracted from stored leaf samples with the Plant RNA Extraction Kit (Tiangen, Beijing, China) based on provided directions, after which a TIANScript First Strand cDNA Synthesis Kit (Tiangen, Beijing, China) was used to prepare cDNA. A CFX96 Real-Time PCR Detection System (Bio-Rad, Hercules, CA, USA) and SYBR Premix ExTaq (TaKaRa, Dalian, China) were then utilized for qPCR analyses using appropriate primers constructed with the RealTime qPCR Assay tool (https://sg.idtdna.com/scitools/Applications/RealTimePCR/ accessed on 6 December 2022). PdPP2A genes served as the internal reference controls, and primers used for this study are compiled in Appendix A. The 2ΔΔCt method was used to assess the relative gene expression, and samples were analyzed in triplicate. Results were compared with one-way ANOVAs. 

## 3. Results

### 3.1. Identification of DREB TF Genes

Whole-genome sequencing results from wild almond and sweet almond plants were used to identify *DREB* TF-encoding genes based on the presence of a conserved AP2/ERF domain, with the protein sequences for the identified *DREB*s being shown in Appendix A. In total, 46 and 51 *DREB* genes were identified in the wild almond and sweet almond genomes, respectively. While these genes were distributed across all six chromosomes in wild almond, they were not present on chromosomes 4 or 7 in sweet almond. These genes were designated *PnaDREB1*–*PnaDREB46* according to their chromosomal locations (Figure 1), which were chromosomes 1, 2, 3, 5, 6, and 8, respectively, containing 12 (26.1%), 5 (10.9%), 4 (8.7%), 6 (13.0%), 6 (13.0%), and 13 (28.3%) *PnaDREB* genes, some of which were present in clusters. These uneven *PnaDREB* chromosomal distribution patterns highlight the complexity and diversity of this *DREB* gene family. 

These 46 *PnaDREB*s exhibited predicted sizes ranging from 166 (*PnaDREB15*) to 1273 (*PnaDREB12*) amino acids, with an average molecular weight of 32.14 kDa. The isoelectric point (pI) values for these *PnaDREB*s also varied widely from 4.57 (*PnaDREB18*) to 9.39 (*PnaDREB5*), and 20 and 27 of these *PnaDREB*s, respectively, exhibited cytosolic and nuclear localization patterns (Appendix A). 

### 3.2. PnaDREB Gene Phylogenetic Analyses

Phylogenetic relationships among the *PnaDREB* TF family members and the potential functional characteristics of these proteins were next explored by constructing a phylogenetic tree based upon the full-length DREB protein sequences from *A. thaliana* (54 sequences), wild *P. nana* (46 sequences), *P. dulcis* cultivar ‘Nonpareil’ (51 sequences), *P. avium* (57 sequences), *P. mume* (49 sequences), *P. persica* (53 sequences), and *P. armeniaca* (50 sequences). These DREB proteins exhibited high levels of sequence diversity, and were separated into two broad groups (A and B) as well as five subgroups (Aα, Aβ, Aγ, Bα, Bβ) (Figure 2). Clear differences in the phylogenetic distributions of *DREB*s from different species were observed, with, for example, 15, five, and six *DREB*s from *A. thaliana*, sweet almond, and wild almond plants, respectively, being assigned to the Aβ group. Phylogenetic analyses also revealed a high degree of *DREB* sequence similarity between the sweet and wild almond plants, as confirmed through sequence alignment and collinearity analysis (Appendix A, Figure 2). The nonsynonymous to synonymous mutation rate (Ka/Ks) was assessed as a metric for selective pressure, driving the evolution of this *DREB* gene family(Appendix A). For 38 orthologous gene pairs identified when comparing the sweet almond and wild almond genomes, the Ka/Ks of most gene pairs was <1, while the Ka/Ks for *PnaDREB12*-XP_034226506.1 was >1, suggesting that this gene evolved under purifying selection following the divergence between wild and sweet almond plants, consistent with the potential adaptive evolution of XP_034226506.1.

*DREB* genes clustered within a single group largely exhibited similar functional roles due to co-adaptations and relationships among these genes. For group Aα, homologous genes in the *Arabidopsis* CBF subgroup were chosen to construct a phylogenetic tree, revealing that there were pronounced differences in the numbers and levels of the sequence similarity of CBF homologous genes across species. To better clarify the phylogenetic relationships among these CBF genes, homologous genes from four Prunus species were selected and used to incorporate a phylogenetic tree along with *Arabidopsis*, sweet almond, and wild almond homologs for these genes (Figure 2). Clear differences in the CBF gene numbers were evident across species, with only five *CBF* genes in *P_armeniaca*, while *P. mume* and *P. dulcis* cultivar ‘Nonpareil’ both harbored eight *CBF* genes. 

### 3.3. Structural Analyses of PnaDREB Genes and Proteins

To additionally assess the functions and characteristics of these *PnaDREB*s, conserved motifs and intron/exon positioning were next assessed. The majority of these genes either exhibited no introns (*PnaDREB33*) or a single intron (*PnaDREB12*), whereas *PnaDREB44* and *PnaDREB12* harbored 2 and 21 introns, respectively (Figure 3).

To more fully characterize the diverse structural characteristics of proteins in the *PnaDREB* family, the MEME program was used to identify specific conserved motifs. In total, 10 such conserved motifs were identified among these 46 *PnaDREB*s (Figure 4). Similar motif patterns were observed within each of the established *PnaDREB* subgroups, consistent with their similar functional roles. Motifs 1–4 were conserved AP2 domain motifs that were present in all of these *PnaDREB*s, while motifs 6 and 5 were unique to the *PnaDREB* A and Aα subfamilies, respectively. These *PnaDREB*s thus harbor conserved characteristics consistent with those of the *DREB* family TFs, indicating that they are likely to play related functional roles. 

### 3.4. Identification of Cis-Regulatory Elements and miRNA Target Sites Associated with PnaDREB Genes

To better understand the mechanisms that may regulate the expression of genes in the *PnaDREB* family, the 2000 bp regions preceding the start codon for each member of this gene family were subjected to promoter analyses that ultimately identified *cis*-regulatory elements associated with light, low-temperature, and drought stress responsivity as well as hormone-responsive elements(Figure 5). These included regulatory elements responsive to methyl jasmonate, gibberellin, auxin, abscisic acid, and salicylic acid signaling. Based on orthologous gene analyses in sweet almonds, light-, ABA-, and MeJA-responsive elements were found to be the most abundant in these analyzed *PnaDREB* promoter regions. Of note, 17 of these *PnaDREB* genes (*PnaDREB1/3/9/12/16/18/19/20/25/27/31/32/35/36/37/38/40*) harbored low temperature-responsive elements, suggesting a need to further study these genes in the context of cold stress. 

To evaluate the potential roles of microRNAs (miRNAs) as regulators of *PnaDREB* expression, a miRNA target site prediction analysis was performed. This approach suggested that 79 miRNAs may regulate these *PnaDREB*s (Appendix A), with *PnaDREB2*, for example, being subject to regulation by *miR-156*, *miR-774*, *miR-854*, and *miR-5638*.

### 3.5. PnaDREB Interaction Analyses

The STRING database (v 10.5, https://cn.string-db.org/cgi/network accessed on 7 December 2022) was next used to predict potential functional interactions for these *PnaDREB* proteins based on their homologs identified in *Arabidopsis* (Figure 6). *PnaDREB36* and *PnaDREB40* exhibited a high degree of homology to *Arabidopsis* CBF1, a key inducer of cold-regulated gene expression that enhances cold tolerance. Interaction analyses suggested close relationships between *CBF1* and *CBF4*, *ABI4*, *DREB2B*, *DREB2C*, *RAP2.9*, and *RAP2.4*, which are homologous to the *P. nana* proteins *PnaDREB37*, *PnaDREB25*, *PnaDREB26*, *PnaDREB24*, *PnaDREB5*, *PnaDREB9*, respectively.

The nodes with different colors represent different proteins and the lines with different color represent different types of evidence for the interaction. Light blue lines show known interactions from curated databases, rose red lines show known interactions experimentally determined, blue lines show predicted interactions of gene co-occurrence, black lines show interactions of co-expression, and yellow lines show interactions of text mining.

### 3.6. Analysis of Cold Stress-Related Changes in PnaDREB Expression Patterns

The potential roles of specific *PnaDREB*s as regulators of cold stress responses were assessed by selecting 15 of these genes and evaluating their responses to a range of different temperatures (25 °C, 5 °C, 0 °C, −5 °C, and −10 °C). These genes included eight that were homologous to *Arabidopsis* CBFs (*PnaDREB22/31/36/38/40/41/42/43*) as well as seven harboring low-temperature *cis*-responsive elements within their promoter regions (*PnaDREB1/3/9/12/16/18/19/27*).

The majority of the analyzed *PnaDREB*s exhibited varying degrees of induction in response to cold treatment (Figure 7), with some exhibiting similar induction patterns in both wild and sweet almond seedlings. *PnaDREB3*, *PnaDREB27*, *PnaDREB31*, *PnaDREB38*, *PnaDREB41*, *PnaDREB42*, and *PnaDREB43* were all upregulated in response to cold stress in both of these species, whereas *PnaDREB9*, *PnaDREB18*, and *PnaDREB22* were downregulated, suggesting that these genes may play roles in cold stress responses. Notably, *PnaDREB27*, *PnaDREB41*, *PnaDREB42*, and *PnaDREB43* were upregulated more than 10-fold relative to the control (25 °C) conditions. The expression levels of *PnaDREB27*, *PnaDREB41*, and *PnaDREB42* were greater in sweet almond relative to wild almond at different tested temperatures, whereas *PnaDREB12*, *PNADREB16*, *PnaDREB31*, *PnaDREB38*, and *PnaDREB39* were expressed at significantly higher levels in wild almond. Strikingly, *PnaDREB16*, which were respective members of subgroups Bα, was upregulated in wild almond plants but downregulated in sweet almond plants in response to cold treatment. These differential expression patterns may belie differences in the cold stress tolerance of these two almond varieties. *PnaDREB1* was downregulated in sweet almond plants, but was first downregulated and then upregulated at −10 °C in wild almonds (Figure 7a). *PnaDREB12* was significantly upregulated in wild almond leaves at −5 °C and −10 °C, whereas it was downregulated in sweet almond leaves at 5 °C and −10 °C (Figure 7b). *PnaDREB19* expression levels in the leaves of wild almond plants also declined in response to cold treatment but rose significantly in sweet almond plants at both −5 °C and −10 °C (Figure 7).

## 4. Discussion

*P. nana* is a wild species that is related to the more widely cultivated sweet almond species. *P. nana* trees have evolved a diverse range of adaptive genetic mechanisms that enable them to tolerate stressors to which they are exposed in their harsh habitats [37]. According to the different uses of almonds, almonds are divided into many varieties. Among them, sweet almonds are widely cultivated for their edible functions, but their distribution in the wild is limited due to their suitability for growing in warm and arid regions. Both drought and cold stress resistance-related genes are invaluable genetic resources that have the potential to aid in efforts to improve the hardiness of cultivated almond varieties. AP2/ERF family TFs are important regulators of plant growth and development [38,39]. *DREB* transcription factors play an important role in plant response to cold stress signaling pathways [40,41]. However, there are few studies on the response of *P. nana* to cold stress at the molecular level. This study is the first publicly published analysis and identification of *DREB* transcription factors through genomic data, providing a basis for a better understanding of the functions of the *DREB* gene family of *P. nana* under cold stress. Accordingly, this study conducted a whole genome analysis of *P. nana* in an effort to identify target genes with the potential to aid in the future breeding of almond cultivars exhibiting superior cold resistance.

In the present study, we named these 46 *PnaDREB* genes from *PnaDREB1* to *PnaDREB46* based on their order on the chromosomes and classified them into five subgroups according to the phylogenetic relationships with *A. thaliana*. The results showed that the number of *DREB* genes in A was similar to that of reported plants such as *A*. *thaliana* 57 *DREB*s; *Triticum aestivum* 57 *DREB*s [42]; *Phyllostachys edulis* 47 *DREB*s [43]; *Sorghum bicolor* 52 *DREB*s [44]; *Cicer arietinum* 43 *DREB*s [45]. The above data indicate that the *DREB* gene family of A is highly conserved, which may be related to gene duplication during speciation and evolution. According to the analysis results of the phylogenetic tree, among the five subgroups, the proportion of subgroups was the highest at 45.7% and 32.6%, respectively, in Aγ and Bβ. The subgroup had the least *PnaDREB* genes, with only one and four, respectively. ICE1 belongs to the *bHLH* transcription factor family. Under cold stress, it activates the expression of *CBF*s genes, ultimately enhancing the cold resistance of plants by activating cold resistant proteins or ROS scavenging systems [46]. *CBF*s are located in the *DREB* family Aα subgroup, and it has been reported that Aα family members are sensitive to cold stress, almost all of which can be induced by cold stress, and can regulate the expression of cold stress-related genes [47,48]. In summary, the *DREB* family of *P. nana* may have the ability to cope with cold stress, laying a foundation for subsequent experiments.

*PnaDREB* genes have different numbers of introns, and most of the identified *PnaDREB* genes contain 0 or 1 intron, although *PnaDREB12* contains 21 introns. In the description of the presence of *PnaDREB12* Aα, there were some mutations. It is worth noting some *PnaDREB* genes, especially B β. The *PnaDREB* gene in group B is generally deficient in introns, which may shorten the post transcriptional process to respond immediately to abiotic stresses [49]. In this study, in Aα, the large number of introns in group I suggests that the molecular structure of group I of the *DREB* gene may be relatively conservative during evolution, which is conducive to evolution caused by protein diversity [50,51]. It seems that the key function of the *DREB* gene family in A is similar to that of Aα. Group A is closely related, and similar results have been found in *Solanum torvum* L. [52]. With the exception of the 1273 amino acids in the *PnaDREB12* protein, the other *PnaDREB*s identified herein were predicted to range from 166 (*PnaDREB15*) to 515 (*PnaDREB16*) amino acids in size, with an average length of 295 amino acids. Molecular weight (17.91–58.10 kDa) and pI (4.57–9.39) values for these proteins were also relatively similar to those for other species including pepper (*Capsicum annuum* L.; 12.13–59.27 kDa, 4.6–10.64 pI) [53], suggesting that the characteristics and functions of these TFs are conserved across species to some degree. These results thus provide a foundation for future efforts to explore the functions of *P. nana DREB*s.

According to the prediction of cis-acting regulatory elements, *PnaDREB*s may play a regulatory role in various biological processes in cotton. The low-temperature responsive (LTR) element (CCGAAA) may be an important mediator of the ability of *PnaDREB* proteins to respond to cold stress, given that it was present in the promoter regions of seven *PnaDREB*s. This LTR sequence has been shown to be an essential *cis*-regulatory element within heat-shock protein promoter regions in both *Ulva prolifera* and *Hordeum vulgare* [9,54]. Sweet almonds were the a control for *P. nana*, B α subgroup *PnaDREB16*, was both upregulated in wild almond plants and downregulated in sweet almond plants. *PnaDREB3*, *PnaDREB12*, *PnaDREB16*, *PnaDREB27*, *PnaDREB31*, and *PnaDREB38* were expressed at significantly higher levels in wild almond seedlings exposed to cold stress, suggesting that they may be important regulators of cold stress responses worthy of further study. After cold treatment, the expression levels of *PnaDREB9*, *PnaDREB18*, and *PnaDREB22* in the selected 15 putative *PnaDREB* genes were downregulated. During cold treatment, we found that the expression levels of *PnaDREB27*, *PnaDREB41*, and *PnaDREB42* in sweet almonds were higher than those in wild almonds. This suggests that some *PnaDREB* transcription factors may play a negative regulatory role. In summary, our analysis adds evidence that *DREB* can assist plants in coping with cold stress.

## 5. Conclusions

This study is the first genome-level description of the *DREB* gene family of *P. nana*. We identified 46 *DREB* genes in *P. nana*, and all of them were located on six chromosomes. In addition, based on phylogenetic relationships, *PnaDREB*s were classified into three groups. The motifs were highly conserved in the same subgroup. Meanwhile, cis-acting regulatory elements indicate that *PnaDREB*s play an important role in coping with cold stress. *PnaDREB* interaction analyses showed that *PnaDREB36* and *PnaDREB40* exhibited a high degree of homology to *Arabidopsis CBF1*. The qPCR results demonstrated that among the 15 selected *PnaDREB* genes, seven genes were significantly upregulated in both plant species, and three genes were significantly downregulated, of which the Bα and Aα members of the subgroup showed the opposite behavior when the two plants responded to cold treatment. 

In summary, the results of this study not only provide clues for future analysis of the mechanism of *PnaDREB* response to cold stress, but also in expanding our understanding of the *DREB* family in plants.

## Figures and Tables

**Figure 1 genes-14-00811-f001:**
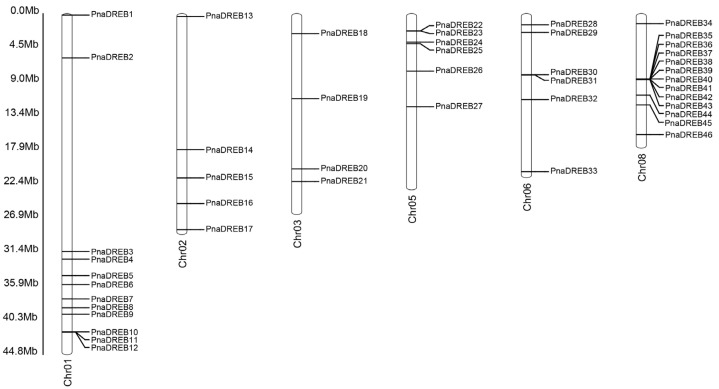
*PnaDREB* gene distributions across six *P. dulcis* chromosomes.

**Figure 2 genes-14-00811-f002:**
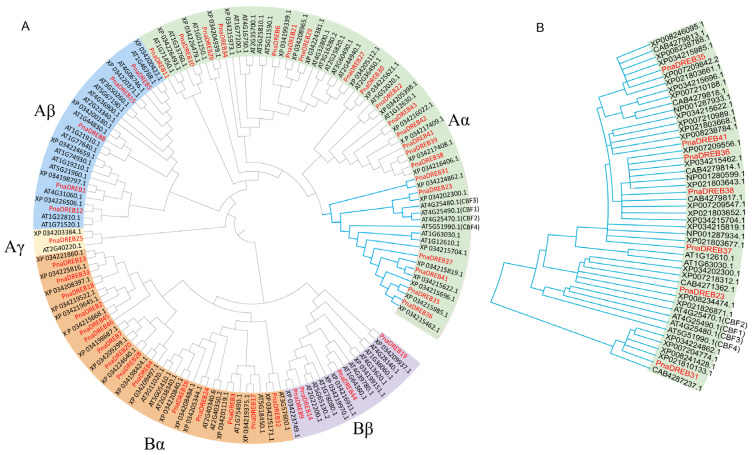
Phylogenetic analysis of the *DREB* family genes. (**A**) *DREB* family genes from *A. thaliana*, wild *P. nana*, and *P. dulcis var.* were subjected to a phylogenetic analysis. (**B**) Phylogenetic analysis of homologous *CBF* genes from *A. thaliana*, *P. nana*, *P. dulcis* cultivar ‘Nonpareil’, *P. avium*, *P. mume*, *P. persica*, and *P. armeniaca*. MEGA 6.0 was used to generate a maximum likelihood (ML) phylogenetic tree with γ-distributed rates and Jones–Taylor–Thornton. The resultant phylogenetic tree was separated into two large subgroups (**A** and **B**) and five subgroups are represented with different colors. Red indicates *Dreb* genes encoded by *P. dulcis*. The initial letter indicates different species such as XP_021: *P. avium*; XP_034: *P. dulcis* cultivar ‘Nonpareil’; XP_008: *P. mume*; XP_007: *P. persica*; CAB: *P_armeniaca*.

**Figure 3 genes-14-00811-f003:**
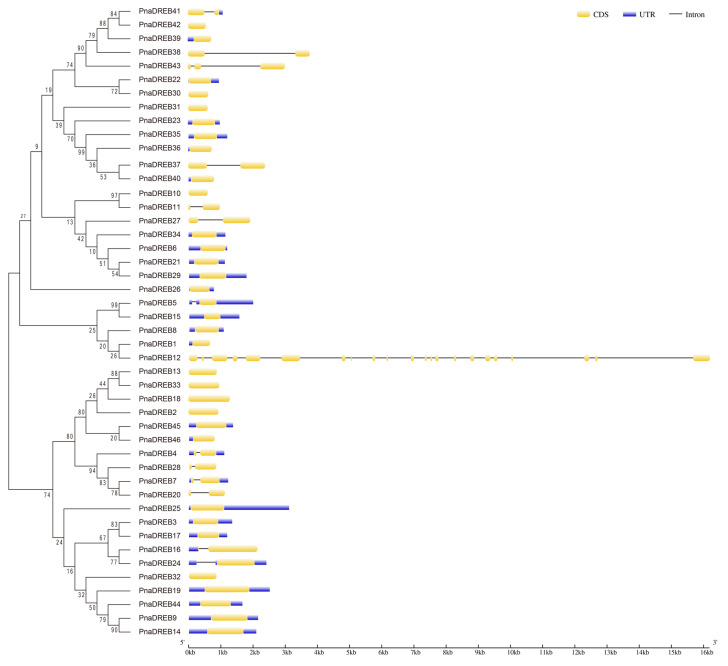
*PnaDREB* gene structure analyses. Exons, introns, and 5′/3′-untranslated regions are represented by the yellow box, black line, and pink box, respectively.

**Figure 4 genes-14-00811-f004:**
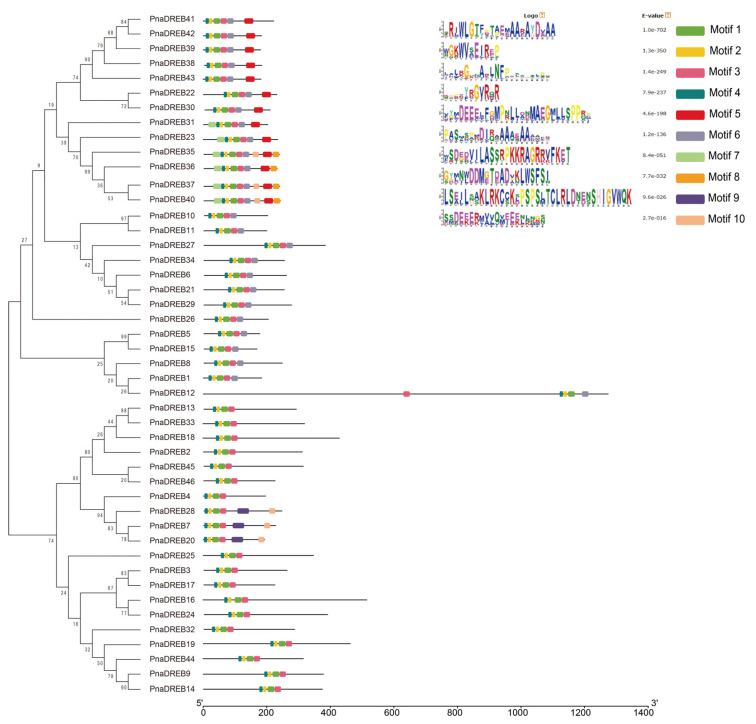
*PnaDREB* gene motif analyses. Different conserved motifs are represented with boxes colored based on motif identity. In total, 10 conserved motifs were identified.

**Figure 5 genes-14-00811-f005:**
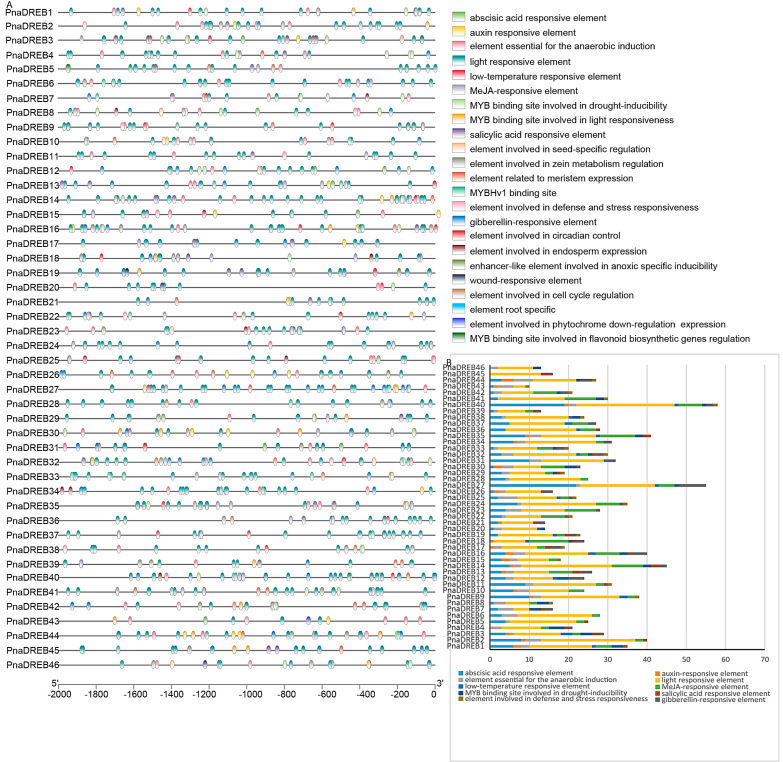
*Cis*-regulatory element predictive analyses of the *PnaDREB* promoter regions. Different colored boxes represent regulatory elements. (**A**) Phylogenetic tree with colored blocks of position and type of elements. (**B**) A bar chart displays the total number of elements identified and the number of each element in a specific color.

**Figure 6 genes-14-00811-f006:**
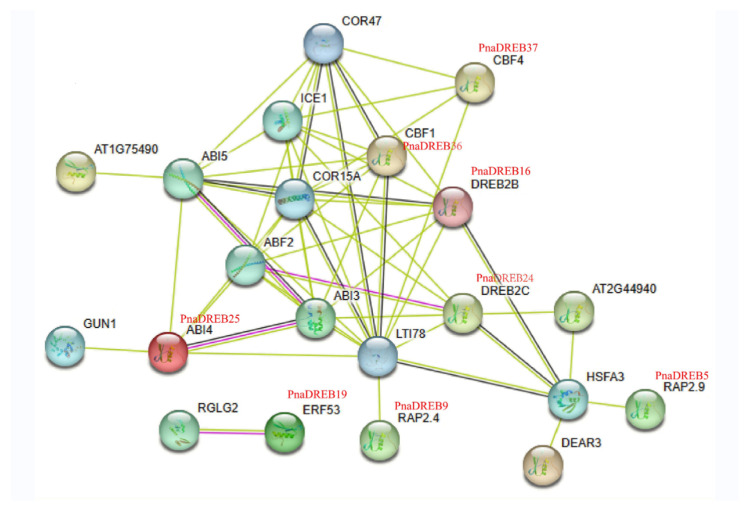
Interaction network for *PnaDREB*s based on their close homologs in *A. thaliana*.

**Figure 7 genes-14-00811-f007:**
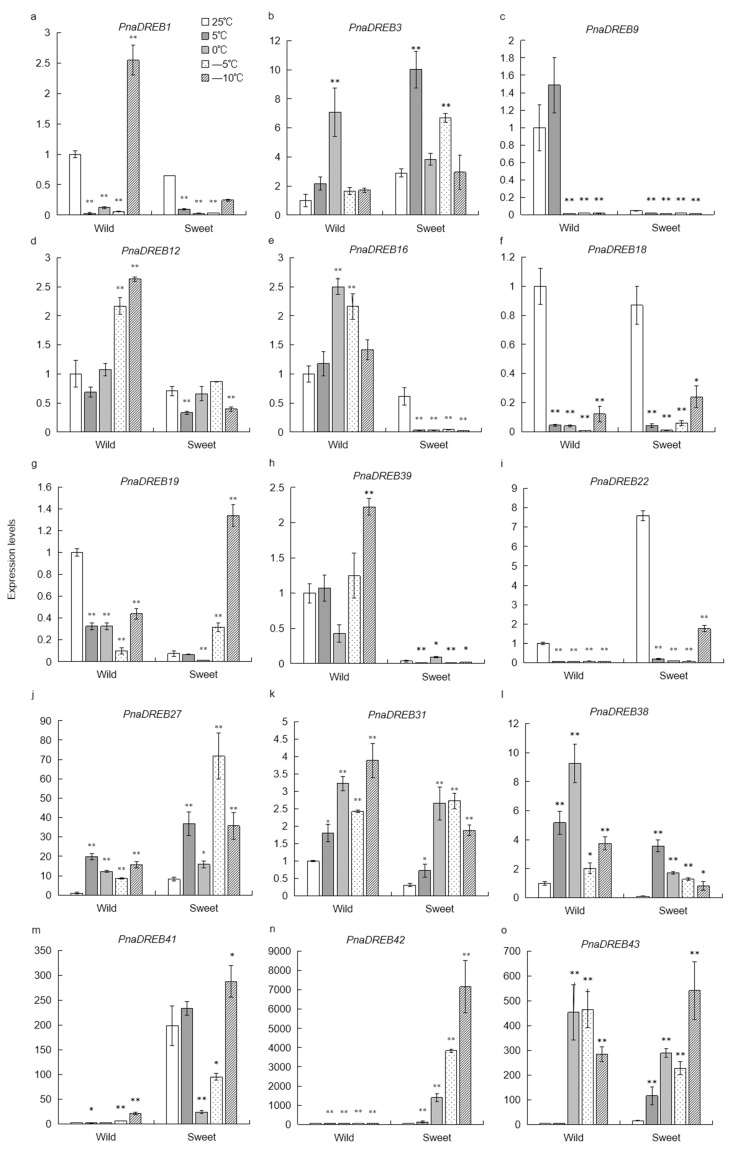
qPCR-based validation of *PnaDREB* expression patterns in plants subjected to low-temperature stress (**a**–**o**). Seedlings were subjected to 2 h treatment at 25 °C, 5 °C, 0 °C, −5 °C, or −10 °C. Mean expression levels are based on three replicate samples, and are presented with standard deviation values. * and ** indicate significant differences between the experimental treatments and control treatment (according to Student’s *t*-test) at *p* < 0.05 and *p* < 0.01, respectively.

## Data Availability

Data supporting the present study are available from the corresponding author upon reasonable request.

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
