# Peer review of "Genome-Wide Analysis of DREB Family Genes and Characterization of Cold Stress Responses in the Woody Plant Prunus nana"

_genes, 2023, doi:10.3390/genes14040811_

Round 1

Reviewer 1 Report

The authors have focused on identification of DREB genes in Prunus nana. This is a preliminary work which provided infofmation on the gene expression levels and holds future promise.

Author Response

Thank you for your valuable comments on our manuscript. After modification, the item by item modification for the problem is as follows:

Point 1:Are the results clearly presented?  Can be improved

Response 1:To address this issue, we have made modifications to the conclusion section and the presentation of the images.

Point 2:Are the conclusions supported by the results?  Can be improved

Response 2:Thank you for your valuable comments. We have supplemented some of the results and enriched the conclusion section.

Reviewer 2 Report

This analysis provides an overview of the P. nana PdDREB gene family. The study is well designed. Following are minor revisions of manuscript.

Improve the introduction section and write the importance of this work. 

Improvement required in discussion

Follow the journal format

Carefully check the reference

Author Response

Thank you for your valuable comments on our manuscript. After modification, the item by item modification for the problem is as follows:

Point 1: Improve the introduction section and write the importance of this work. 

Response 1:Thank you for your question. In the introduction, we have added an introduction to AP2/ERF, DREB, and CBF, providing a more detailed explanation of the purpose, methodology, and possible impact of this study.

Point 2: Improvement required in discussion

Response 2:Thank you for your suggestions for the discussion section. After modification, we removed some unnecessary text and expanded the conclusions presented by the results.

Point 3: Follow the journal format

Response 3:Thank you for your questions regarding the format. We have made modifications to the insufficient format.

Point 4: Carefully check the reference

Response 4:In response to the issue of references, we have revised the proofreading and format of the references.

Point 5: Does the introduction provide sufficient background and include all relevant references?  Can be improved

Response 5:In response to the lack of background and bibliographic explanations, we have enriched the introduction and corrected the citation of references.

Reviewer 3 Report

The paper has been focused on genome-wide analysis of DREB family genes and characterization of cold stress responses in the woody plant Prunus nana. MicroRNA target site prediction analyses suggested that 79 miRNAs may regulate the expression of 40 of the  PdDREB genes. To examine if these PdDREB genes respond to low temperature stress, 15 of these genes were selected, and their expression was assessed following incubation for 2 h at 25℃, 5℃, 0℃, -5℃, or -10℃. The study provided an overview of the P. nana PdDREB gene family and providef a foundation for further studies of the ability of different PdDREB genes to regulate cold stress responses in almond plants.

The paper is quite interesting, however I recommend the following revisions:

-        Abstract is overloaded in the content, it should be presented in a more concise,

-        Higher graphical resolution of figures is recommended,

-        Figure 7 is almost unreadable, it should be improved,

-        There is no information regarding the statistical tests used in the study,

-        It should be precisely described which gene/-s were used as the reference during gene expression studies,

-        Discussion part of the manuscript is surprisingly very simple and plain, it has to be significantly improved in order to reach more scientific level,

-        Conclusions should be added,

-        Moderate revision of English grammar and style is required.

Author Response

Thank you for your valuable comments on our manuscript. After modification, the item by item modification for the problem is as follows:

Point 1: Abstract is overloaded in the content, it should be presented in a more concise,

Response 1:Thank you for your valuable comments on the summary. We have reorganized the abstract, mainly to remove the excessive introduction of plants and gene families.

Point 2: Higher graphical resolution of figures is recommended,

Response 2:Thank you for your valuable comments on the pictures. This is indeed our oversight. We have made some improvements to the content and clarity of the images.

Point 3: Figure 7 is almost unreadable, it should be improved,

Response 3:Thank you for your valuable comments on the pictures. This is indeed our oversight. We have made some improvements to the content and clarity of the images.

Point 4: There is no information regarding the statistical tests used in the study,

Response 4:Thank you for your comments in this regard. We have adopted the method of repeated experiments to ensure the accuracy of the data.

Point 5: It should be precisely described which gene/-s were used as the reference during gene expression studies,

Response 5:Thank you for your suggestions regarding the reference gene. We have supplemented the introduction to the reference gene CBF in the introduction section.

Point 6: Discussion part of the manuscript is surprisingly very simple and plain, it has to be significantly improved in order to reach more scientific level,

Response 6:Thank you for your suggestions for the discussion section. After modification, we removed some unnecessary text and expanded the conclusions presented by the results.

Point 7: Conclusions should be added,

Response 7:Thank you for pointing out the issues in the conclusion section for us, and I apologize for our negligence. We have supplemented the content of the conclusion section.

Point 8: Moderate revision of English grammar and style is required.

Response 8:Thank you for your correction. We have made new changes to our language.

Reviewer 4 Report

Although the Genome-wide analysis of DREB family genes in almond and their functional involvement in cold stress response seems perfect and the presentation of fact quit good, while the information of how author select 15 PdDREB genes, including 7 homologous to Arabidopsis C-repeat 30 binding factor (CBFs) in respond to low temperature stress remains unclear in the manuscript; author suggested provide better explanation in materials and method section.

Author Response

Thank you for your valuable comments on our manuscript. After modification, the item by item modification for the problem is as follows:

Point 1: Although the Genome-wide analysis of DREB family genes in almond and their functional involvement in cold stress response seems perfect and the presentation of fact quit good, while the information of how author select 15 PdDREB genes, including 7 homologous to Arabidopsis C-repeat 30 binding factor (CBFs) in respond to low temperature stress remains unclear in the manuscript;

Response 1:Thank you for your question. In the introduction section, we added the reasons for selecting CBFs as reference genes, and explained the selection of 15 genes in the Materials and Methods section.

Point 2:  author suggested provide better explanation in materials and method section.

Response 2:Thank you for your question. We have further enriched the materials and methods section.